# The Influence of Printing Layer Thickness and Orientation on the Mechanical Properties of DLP 3D-Printed Dental Resin

**DOI:** 10.3390/polym15051113

**Published:** 2023-02-23

**Authors:** Andrei Zoltan Farkas, Sergiu-Valentin Galatanu, Riham Nagib

**Affiliations:** 1Department of Mechatronics, Polytechnic University of Timisoara, 1 Mihai Viteazul Blvd., 300222 Timisoara, Romania; 2Department of Mechanics and Strength of Materials, Politehnica University of Timisoara, 1 Mihai Viteazu Blvd., 300222 Timisoara, Romania; 3Orthodontics Research Center ‘ORTHO CENTER’, Department of Orthodontics, “Victor Babes” University of Medicine and Pharmacy Timisoara, Eftimie Murgu Sq. 2, 300041 Timisoara, Romania

**Keywords:** C&B Micro Filled Hybrid, dental resin, 3D printing, tensile test, compression test

## Abstract

Technological advances are closely related to the development of new materials and their processing and manufacturing technologies. In the dental field, the high complexity of the geometrical designs of crowns, bridges and other applications of digital light processing 3D-printable biocompatible resins is the reason for the need for a deep understanding of the mechanical proprieties and behavior of these materials. The aim of the present study is to assess the influence of printing layer direction and thickness on the tensile and compression proprieties of a DLP 3D-printable dental resin. Using the NextDent C&B Micro-Filled Hybrid (MFH), 36 specimens (24 for tensile strength testing, 12 for compression testing) were printed at different layer angulations (0°, 45° and 90°) and layer thicknesses (0.1 mm and 0.05 mm). Brittle behavior was observed in all specimens regardless of the direction of printing and layer thickness for the tensile specimens. The highest tensile values were obtained for specimens printed with a layer thickness of 0.05 mm. In conclusion, both printing layer direction and thickness influence mechanical proprieties and can be used to alter the materials’ characteristics and make the final printed product more suitable for its intended purposes.

## 1. Introduction

Technological advances are closely related to the development of new materials and their processing and manufacturing technologies [1]. 

Three-dimensional (3D) printing is a technology that has enabled the manufacturing of complex structures, with comparatively short times and less material consumption when compared to classical manufacturing technologies [2]. In recent years, in more and more fields, an increasing emphasis has been placed on additive manufacturing techniques [3]. More formally known as additive manufacturing (AM), 3D printing, besides being adopted for rapid prototyping, is also used for rapid manufacturing and shows great potential for applications in aerospace engineering, biomedical engineering, and conceptual model preparation. The 3D printing market is constantly on the rise, and it is projected that it will continue to thrive in the future [4].

Digital light processing (DLP) is a vat-photopolymerization-based 3D printing technology where full layers of photosensitive resin are irradiated and cured with projected ultraviolet (UV) light. The final product is a three-dimensional part created layer by layer. Recent breakthroughs in polymer chemistry have led to a growing number of UV-curable elastomeric photo resins developed exclusively for VAT photopolymerization [5,6,7]. 

DLP 3D printing technology is used to produce fine components due to its high precision and efficiency, being advantageous for fabricating complex structures used in various fields, including dentistry [8]. The novel elastomeric photo resins combined with the practical manufacturing advantages of DLP are suitable for applications requiring different degrees of flexibility, stretchability, conformability, or stiffness [9].

As additive manufacturing of polymeric materials is becoming more prevalent throughout the dental field and other industries, it is important to ensure that 3D-printed parts are able to withstand environmental and mechanical stresses that occur when they are in use [10]. 

Recent advances brought by the evolution of photopolymerizable resins and DLP technologies have also had a considerable impact on dental medicine [11,12], with 3D-printed resins being used in a wide range of clinical applications [13,14,15,16,17]. Although these resins are often used in clinical scenarios, some of them lack information and experimental studies related to their mechanical properties. 

Literature on 3D-printed resins, among other subjects, deals with the influence of various parameters on fatigue crack propagation and material failure or fatigue behavior [10], or the strain-rate-dependent mechanical properties of 3D-printed polymer materials, such as polyurethane acrylate resin [11]. Research articles explore the possibility of heat treatment in order to obtain a strengthening effect on the printed objects [9], or the impact of changing the process parameters on the tensile properties of DLP resins [8]. Experiments have been conducted to determine the mechanical properties of 3D-printed parts fabricated using DLP and the effect of print orientation angle on the friction and wear properties. Experiments that assess the effect of post-processing with heating and UV light to achieve a better cured surface finish for the tensile, friction, and wear tests [18], and experiments regarding the printing layer thickness and curing exposure time effects on the mechanical properties of dental composite resins [19] have also been undertaken. Researchers additionally made comparisons between the mechanical properties of DLP-printed resin and graphene/resin composite materials [2], while others investigated the elastic properties of different types of strut-based lattice structures obtained with DLP technologies [20] or managed to combine different monomers in order to obtain self-healing, recyclability, and tailorable mechanical properties [21]. In the field of biomedicine, studies managed to use polylactic acid resin in combination with a DLP 3D printing method that was synthesized to produce hard tissue scaffolds in order to mimic biological structures [22]. The development of new resins or printing techniques is an ongoing process, either by synthesizing and formulating new resin formulas for DLP 3D printers [23] or proposing other printing techniques at different printing temperatures [24].

In this research article, the focus lies on the DLP resin, C&B micro-filled hybrid (MFH) from NextDent (3D Systems, Rock Hill, SC, USA). This resin is a CE certified Class II biocompatible photopolymerizable resin developed for crowns and bridges that is widely used in dentistry [25,26], in dental laboratories and clinics for its intended purpose, but for other applications in the field as well [27]. In the mechanical characterization of 3D-printed polymers, the commonly used ASTM and ISO mechanical test standards have been used by various research groups to test the strength of the 3D-printed parts [28]. In the case of the aforementioned material, the manufacturer specifies the flexural strength in the technical sheet [29] but more data about the mechanical proprieties of the material is needed for a deeper understanding of the clinical behavior of the printed products and also for the finite element method (FEM) analysis of conceptual designs.

Research articles regarding C&B MFH resin cover the impact of post-printing cleaning methods on the degree of conversion, as well as surface and mechanical properties after artificial aging [30], changes in the physicochemical and clinical characteristics of 3D-printed polymers [31], and their conformity to the imposed standardized requirements. The literature also includes studies on the adhesion of yeast fungi and oral bacteria to various types of in vitro samples of polymeric material samples [32], or their cytotoxicity, [33] and even the color and optical properties of 3D-printed resins [26].

Regarding the mechanical properties of the C&B MFH resin, there is some research on the influence on the physical and mechanical properties of double bond conversion during polymerization [34], the wear resistance, hardness, and flexural strength of the resin [35], but there is a knowledge gap when speaking about the tensile or compression strength of the resin, values that are required as material properties in FEM analysis [36]. Certain material properties such as mass density, Poisson’s ratio, and Young’s Modulus, are also required to perform an FEM analysis [37]. Although some characteristics for the material used in the present study, such as the mass density and the Poisson ratio, are found in the literature [38], there are others, for example the Young’s Modulus, that are not. 

The aim of the present study is to determine the influence of printing layer direction on the tensile and compression proprieties of a DLP 3D-printable dental resin and the influence of printing layer thickness on the tensile properties. The article also aims to experimentally determine the mechanical tensile and compressive properties for the NextDent C&B MFH resin, thus obtaining the resin’s Young’s Modulus for the two mechanical tests.

## 2. Materials and Methods

A one-kilogram container of NextDent C&B MFH, color-BL, LOT XG283N23, produced by 3D Systems (Rock Hill, SC, USA) was purchased from Metrodent (Huddersfield, UK), an authorized UK distributor [39].

Specimens to be 3D printed for mechanical testing were modeled according to the standard D638-14 test method for the tensile properties of plastics [40], respectively, according to the standard D695-02a test method for the compressive properties of rigid plastics [41].

The specimens for the tensile and for the compression tests were modeled in CATIA (V5R19, Dassault Systèmes, Vélizy-Villacoublay France) at 0°, 45°, and 90° to the printing tray, resulting in 3 different angles of printing/photopolymerization layer [Figure 1].

The specimens were printed using an “ANYCUBIC Photon Mono X” 3D printer. This digital light processing (DLP) printer is similar to the mid- or higher-priced models available on the market [42] when it comes to the printing quality, but the ANYCUBIC prints with 4K resolution models with a base of up to 270 mm/290 mm and a height of 475 mm [43].

The CAD model was exported in the STL format to be processed and prepared for printing in the dedicated software, Chitubox (V1.8.0, CBD-Tech, Shenzhen, China), to obtain the ‘G’ code necessary for printing the tensile test specimens. 

The printing settings used to print the test specimens are presented in Table 1.

The printed specimens for tensile strength determination were as follows: 12 tensile specimens marked “P1–P12” for a thickness of 0.1 mm and 12 tensile specimens marked with “S1–S12” for a thickness of 0.05 mm. There were 4 specimens for each orientation of the photopolymerization layer, respectively, at 0°, 45° and 90° (Figure 2a,b).

For the compression tests, 12 specimens marked with “1–12” (Figure 2c) were printed. There were 4 specimens for each orientation of the photopolymerization layer, respectively, at 0°, 45° and 90°. All compression specimens were printed with a layer thickness of 0.1 mm. All specimens were printed at once, in order to ensure the same environmental printing conditions. 

After printing, all specimens were rinsed multiple times in an alcohol solution [96%] and treated with UV light according to the specifications given by the resin manufacturer [29], a necessary procedure to obtain a final biocompatible product suitable for clinical use.

The dimensions of all specimens were measured with an electronic caliper according to precision standards (two consecutive measurements were made by the same operator with the same device). All collected data were inserted in spreadsheets (Windows Excel Office 365 software) and descriptive statistics was used to assess printing accuracy and the homogeneity of the study specimens.

Tensile tests were performed on the universal tensile/compression testing machine: Zwick/Roell Z005 (Figure 3a). The universal machine was equipped with two clamping bars, one fixed and one mobile to determine the breaking strength and the longitudinal modulus of elasticity of the tested material. This test was carried out by applying a progressive stretching force F in the direction of the longitudinal axis of the sample, which produces gradual deformation and, finally, breaks the specimens. To determine the elongation of the specimen, an electronic extensometer with a stroke of 10 mm and an initial opening of 30 mm was used. All tests were conducted in the same standardized laboratory conditions to avoid the risk of bias errors.

Setup and data processing were performed using Text Expert software (V2, Zwick Roell Group, Ulm, Germany). The test conditions used for the tensile tests on the Zwick/Roell Z005 [44] universal machine was a speed of 5 mm/min and the maximum value recorded of the force cell was 5 kN;

The compression tests were performed on the universal tensile/compression testing machine: A009-TC100 series 06N/1, with a nominal load of 100 kN, a speed of 5 mm/min, and the computer-controlled actuation force measurement system [45]. 

The test machine consisted of a robust frame with two columns, on which a movable crossbar was mounted. The machine control unit, UDI 16/4, allows the machine command to be tested by means of a computer (PC) and the Soft-TC software (V 2004 Plus, Sen Soft s.r.o., Košice, Slovakia) [Figure 3b]. 

## 3. Results

Measurements of tensile specimens were centralized according to the orientation of the polymerization/printing layers (0°, 45°, and 90°) in Table 2.

Is to be noted that the tensile specimens with a layer thickness of 0.05 recorded values almost five times higher than the SD in the section compared to the ones with a thickness of 0.1 mm.

The SD for the dimensional values of the printed specimens that recorded the highest value was in the 45° layer orientation for both samples printed with a thickness of 0.1 mm and 0.05 mm.

The compression specimens were also centralized according to the orientation of the polymerization/printing layers, as shown in Table 3.

The compression specimens recorded higher SD for the specimens with a printing layer orientation of 45°. The level of SD is considerably bigger than the one registered on the tensile specimens that were printed with the same layer thickness.

Experimental testing results showed the fracture occurred in the calibrated area of the specimens for the tensile specimens with a thickness of 0.1 mm (Figure 4a), and of the 0.05 mm layer thickness specimens (Figure 4b) after the tensile tests.

The fracture of the compression specimens differs with the printing angle (Figure 5c).

Stress–strain curves were plotted after processing the data from the tensile tests. The colors of the stress–strain curves are based on the orientation of the print layer with respect to the printer tray and are marked as follows: yellow for a 0° orientation; red for a 45° orientation and green for a 90° orientation.

After the tests, differences in the direction of crack propagation can be seen between specimens that have different directions of printing layers.

Tensile specimens have the same type of fracture and brittleness, the difference being that those printed with angles of 0° and 45° have the flanks of the fracture area in the shape of a “V”, with the material inside this area missing. The “V” shape can be caused by the gravitational bending effect of the specimens. In the case of those printed at an angle of 90°, the flanks kept an almost parallel direction to each other, the material clearances being very small compared to the other specimens. 

Compression specimens also show considerable differences in the way that their structure failure behaved under compressive pressure; some buckled while others exploded.

Taking into account the force displacement curves, the stress–strain curves were obtained (Figure 5). The stresses were calculated by dividing the force applied in each case by the area of the calibrated section of each specimen tested.

The strain was calculated by dividing the elongation at each step by the initial 30 mm opening of the electronic strain gauge provided with the universal machine.

The tensile tests on the samples printed with a thickness of 0.1 mm registered the maximum stress–strain values among the specimens printed at an angle of 0°, whereas specimens printed at 45° and 90° recorded successively lower values, and the difference was the same for both printing layer thicknesses. The tensile tests on the samples printed with a thickness of 0.05 mm, with print directions of 0° and 90° showed a similarity between the results. The lowest values, for the specimens printed at 0.05 mm, were obtained for those printed at 45°.

The highest stress values were obtained in the case of specimens printed with a layer thickness of 0.05 mm, while the highest strain values were obtained in the case of specimens printed with a layer thickness of 0.1 mm.

The compression test shows that, at the beginning, the deformation is very small and the specimen is elastic. The stress increases linearly with strain, in the elastic regime, and ends when the specimen enters the plastic domain. After the curve passes in the plastic domain, the material passes the yield point into the yielding domain. After yielding, a plateau followed by an increase in stress and an increase in strain can be observed. After reaching the maximum stress, the specimens failed with brittle behavior in most cases.

Following the compression tests, a similar behavior can be observed regardless of the direction of the printing layers. In the yield zone, the shape of the curves is similar for all printing orientations, and the only registered differences are in the maximum stress–strain values registered.

The highest recorded strength was in the case of the specimens printed at an angle of 90° (toward the printer tray), with the 45° and 0° specimens registering successively lower maximum stress values. The strain values followed the same maximum values in the case of the samples printed at 90°, reaching a value of 47%.

The average maximum compression yield stress value obtained for 0° were 85.9 MPa, 98.45 MPa for the 45° specimens, and around 110 MPa for the 90° specimens.

Figure 5 presents the typical stress–strain curves for different printing directions. From this figure, it can be observed that the maximum stress–strain values were obtained between P1, P2, P3 and P4, S1, S2, S3 and S4, and S9, S10, S11 and S12, respectively. The exact values for each specimen and the medium values obtained for each layer orientation are presented in Figure 6a. The maximum compression stress and the yield compression stress by the angle of printing are shown in Figure 6b. 

The average value of the maximum break stress for the 0.1 mm thick layer specimens are 56.8 MPa, registered among the specimens printed at 0°. The average value of the 0.05 mm thick layers are 58.5 MPa, among the specimens printed at 0°, and 58.0 MPa among those printed at 90°.

The average value of the maximum break strain for the 0.1 mm thick layers are 3.147% among the specimens printed at 0° and the average value for the 0.05 mm thick layers are 2.38% among the specimens printed at 0°, respectively 2.41% among those printed at 90°.

The average maximum compression yield stress value obtained for specimens printed at 0° is 85.9 MPa, while it was 98.45 MPa for the 45° specimens and around 110 MPa for the 90° specimens. The peak values obtained are 146.64 MPa for 0°, 228.28 MPa for 45°, and 238.26 for 90°.

The average values of all the measurements for Young’s Modulus and compression modulus determined experimentally are presented in Figure 7.

Young’s Modulus has the highest values for specimens printed with a layer thickness of 0.05 mm at an angle of 45°, registering an average value of 3225.4 MPa. For these specimens, the results were dispersed, showing the homogeneity of the structure, regardless of the printing direction.

The specimens printed with a layer thickness of 0.1 mm have shown a linear trend, increasing the Young’s Modulus from specimens printed at 90° to the those printed at 45° and printed at 0°, where the maximum average value of 2753 MPa is registered. 

However, the compression modulus increases from the specimens printed at 0° toward those printed at 90°, and at the same time the standard deviation decreases with the increasing compression modulus, showing a better photopolymerization of the layers for specimens printed at 90°, which registered the highest average value of 2232.3 MPa. 

If the deviation from the minimum and maximum values registered for each printing layer orientation is to be taken into consideration, it can be observed that the specimens printed at 0° have the highest error prevalence, followed by the specimens printed at an angle of 45°, and finally the ones printed at 90° have the lowest registered deviations. This can be attributed to the potential errors that occurred during the printing process, and thus the high standard deviation from the mean value.

## 4. Discussion

Additive manufacturing DLP technology has received significant interest from dental researchers because of its potential in digital applications. Dental crowns, bridges, and other dental devices [27] can be manufactured using 3D printing technology. That is the reason why the understanding of the mechanical properties of materials used in dental applications is essential, either for comparing them with conventional materials [46], verifying manufacturers claims or simply determining them where they are missing. The current research evaluated the effect of printing layer thickness on the tensile properties and the influence of printing layer orientation on the tensile and compression properties of the C&B MFH resin. 

Numerous factors can alter the mechanical properties of the printed resin, including environmental conditions, post printing conditions, the degree of polymerization, and others [47,48]. Tensile tests undergone in the present study registered the maximum stress–strain values among the specimens printed at an angle of 0°. Specimens printed at 45° and 90˚ recorded successively lower values, with the same difference for both printing layer thicknesses. Similar differences between the orientation of the printing layers have also been reported in other studies on AM technologies [49].

Additionally, the impact and influence of print layer thickness on the mechanical properties for different types of materials is a well-known studied subject [50,51]. 

Data regarding the influence of the thickness on the tensile properties of the C&B MFH resin showed higher stress values in the case of specimens printed with a layer thickness of 0.05 mm compared to those printed with a thicker layer (0.1 mm). The results are similar to experimental results conducted on other 3D-printable resin materials. The majority of studies reported higher tensile proprieties in thinner printing layer products [52,53], as did the present research. One study on dental resins concluded that a greater printing layer thickness is more efficient for polymerization [19].

The highest strain values were obtained in the case of specimens printed with a layer thickness of 0.1 mm. A reason for these results can be the increased size in the cross section of the printed layer, and the excess material leading to a larger plastic zone before the necking effect, before finally breaking [54,55].

Regarding the compression tests, the highest recorded strength was in the case of the specimens with the printing layer at an angle of 90° (toward the printer tray), with the 45° and 0° registering successively lower maximum stress values. A reason for this may be that the “pancake” type placement of the printing layers gives the specimens a better compressive ability [56,57]. 

On the opposite end, the 0° printed specimens measured the lowest values due to the compressing force being orientated along the length of the printed layers, a factor that leads to a buckling effect which significantly reduced the structural stability of the specimens. 

As observed from the differences in standard deviation of the measured specimens, as it is known in the literature [58], build angle influences the accuracy of 3D-printed products. The biggest differences from the 3D models occurred when printing at an angle of 45° regardless of layer thickness. When comparing the differences in deviation between the two thicknesses used, it can be observed that the thinner the layer the bigger the standard deviation from the model’s initial dimensions. These errors can be attributed to two observed phenomena. 

Upon a closer inspection of the tensile specimens, it was observed that those printed at 0°, as well as 45°, have a gravitational bending effect that can lead to an eccentric tensile load that could be the reason for the differences in the direction of the crack propagation of the tensile specimens. The “V” shape can be caused by the gravitational bending effect of the specimens. In the case of those printed at an angle of 90°, the flanks kept an almost parallel direction to each other. The reason why is that the material clearances were very small compared to the other specimens.

Another phenomenon that could be attributed to the printing accuracy is the high standard deviation in the case of the compression samples printed at 0°. In this position, the samples were printed starting from the cylinder’s generator line, forcing the printer to print a different size layer each time in order to obtain the convex shape, leading to printing errors.

The Young’s Modulus for this resin was calculated from the tensile and compression tests for different printing layer orientations and thickness. The values obtained from the tensile and compression tests are in close to the Young’s Modulus that was determined through flexural tests for this resin and can be found in the literature [53].

## 5. Conclusions

Within the limitations of the current study and based on the results, the following conclusions can be drawn:printing layer direction influences the accuracy of the final product when using less thick layers or printing round shapes, and in both cases the 45° orientation leads to the biggest deviations;to increase the tensile strength of this resin after printing, the orientation of the printed layers should be parallel to the 3D printer tray, that is, at an angle of 0°;to increase the tensile strength of this resin after printing, a thinner printing layer is recommended;if better strain properties are needed, printing a thicker printing layer is recommended;Considering that, the maximum yield point is more important when designing a product. To increase the compression strength of dental crowns, bridges, or other dental products that are to be 3D printed with the C&B MFH dental resin, using a printing layer direction of 90° towards the 3D printer’s tray is recommended.

The data obtained experimentally, in addition to guiding technicians in the field of dental medicine regarding the printing settings according to which different values of stress–strain or yield stress can be obtained, could be useful to engineers in order to model and simulate with numerical means, such as the use of software specialized in the finite element method, before printing certain dental products with this resin.

## Figures and Tables

**Figure 1 polymers-15-01113-f001:**
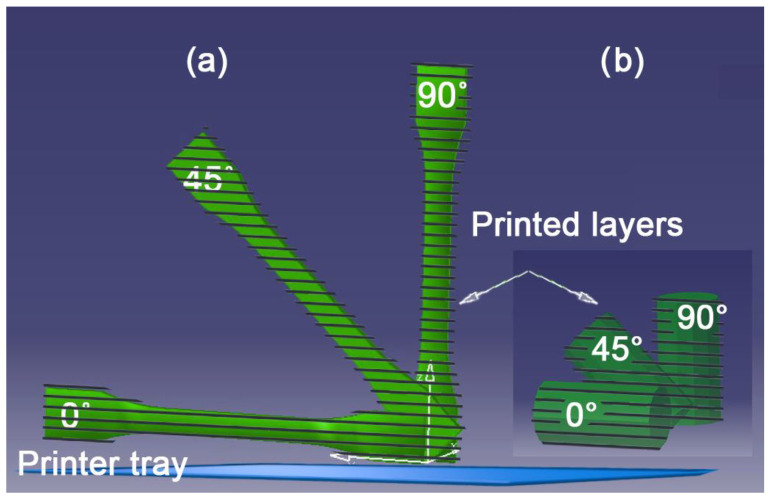
Tensile and compression test specimens modeled according to: (**a**) the D638-14 standard, (**b**) the D695-02 standard, at 0°, 45° and 90°.

**Figure 2 polymers-15-01113-f002:**
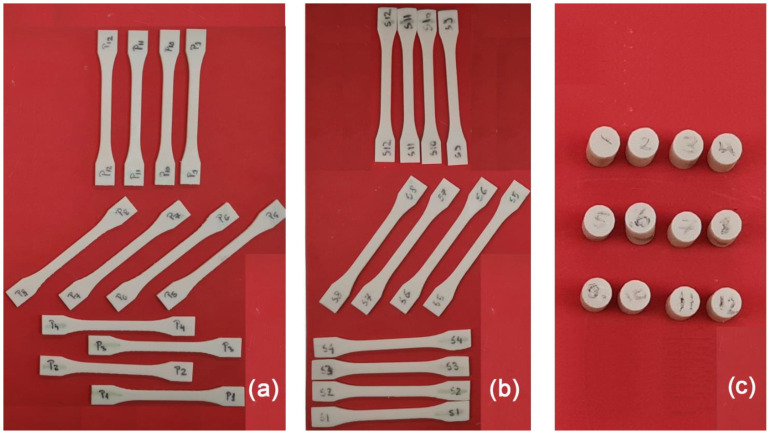
The printed specimens: (**a**) P1–P12 (tensile with 0.1 mm thickness), (**b**) S1–S12 (tensile with 0.05 mm thickness) and (**c**) 1–12 (compression with 0.1 mm thickness).

**Figure 3 polymers-15-01113-f003:**
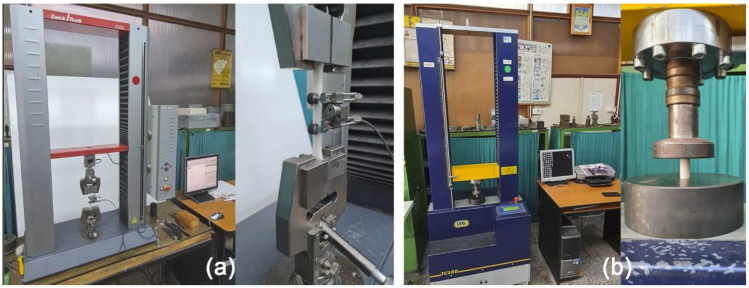
(**a**) Tensile test specimen mounted on the universal tensile/compression Zwick/Roell Z005 testing machine, and (**b**) A009 (TC100) universal testing machine.

**Figure 4 polymers-15-01113-f004:**
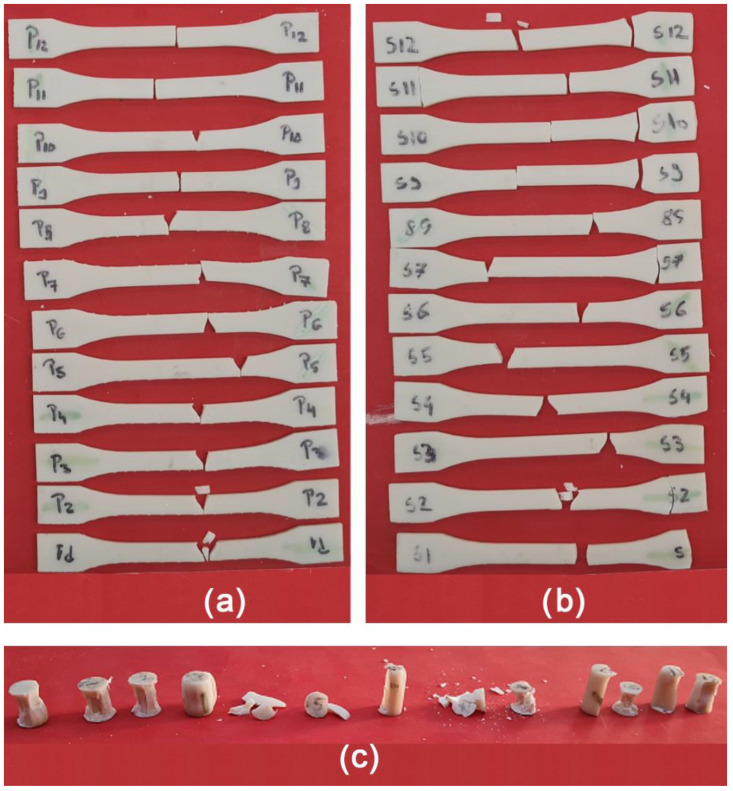
Specimens after the tests: (**a**) tensile specimens with 0.1 mm thickness, (**b**) tensile specimens with 0.05 mm thickness, and (**c**) compression specimens with 0.1 mm thickness.

**Figure 5 polymers-15-01113-f005:**
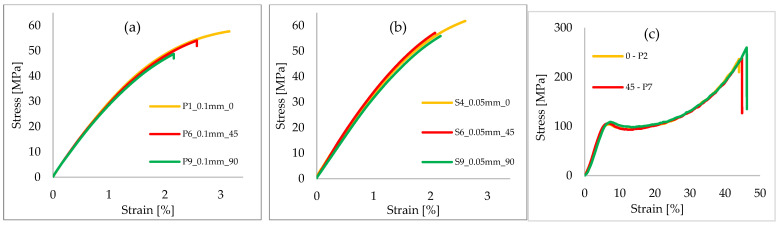
Stress-strain curves for different printing directions (**a**) for the tensile test using specimens with 0.1 mm thickness, (**b**) for the tensile test using specimens with 0.05 mm thickness, and (**c**) for the compression test using specimens with 0.1 mm thickness.

**Figure 6 polymers-15-01113-f006:**
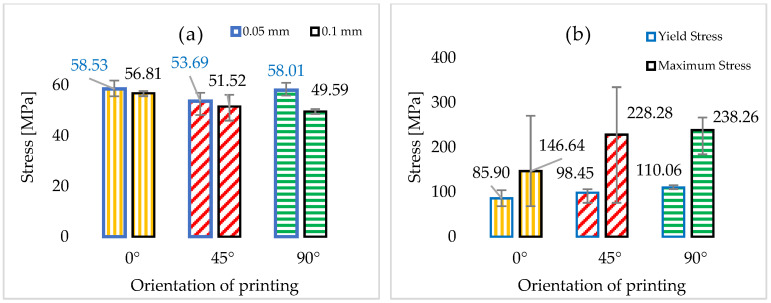
(**a**) Average of the maximum stress for specimens with 0.05 mm and 0.1 mm thickness, and (**b**) average values of the maximum compression stress and yield compression stress by the angle of printing.

**Figure 7 polymers-15-01113-f007:**
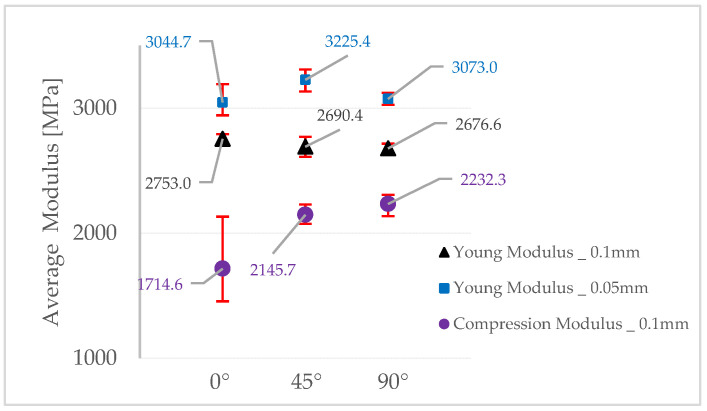
Average of the Young’s/compression modulus.

**Table 1 polymers-15-01113-t001:** Printing settings.

Layer Height:	0.1 [mm]/0.05 [mm]	Bottom Lift Distance:	6 [mm]
Bottom Layer Count:	4 [layers]	Lifting Distance:	6 [mm]
Exposure Time:	3 [s]	Bottom Lift Speed:	120 [mm/min]
Bottom Exposure Time:	15 [s]	Lifting Speed:	120 [mm/min]
Light-off Delay:	8 [s]	Retract Speed:	180 [mm/min]
Bottom Light-off Delay:	8 [s]		

**Table 2 polymers-15-01113-t002:** Main geometric dimensions of the tensile test specimens and the standard deviations (SD) after printing each printing layer orientation.

Specimens	Layer orientation	Measurement 1	Measurement 2	Average Width [mm]	Measurement 1	Measurement 2	Average Thickness [mm]	Cross- Section Area [mm^2^]
		Width [mm]	Width [mm]		Thickness [mm]	Thickness [mm]		
P1	0°	10.10	10.05	10.075	4.00	4.05	4.025	40.55
P2		10.05	10.05	10.05	4.00	4.00	4.00	40.20
P3		10.00	10.00	10.00	4.00	4.00	4.00	40.00
P4		10.05	10.10	10.075	4.00	4.00	4.00	40.30
Min		10	10	10	4	4	4	40
Max		10.1	10.1	10.075	4	4.05	4.025	40.55
SD		0.041	0.041	0.037	0	0.022	0.011	0.230
P5	45°	10.15	10.10	10.125	4.05	4.00	4.025	40.75
P6		10.20	10.20	10.20	4.10	4.00	4.05	41.31
P7		10.20	10.15	10.175	4.10	4.10	4.10	41.71
P8		10.25	10.25	10.25	4.05	4.10	4.075	41.76
Min		10.15	10.1	10.125	4.05	4	4.025	40.75
Max		10.25	10.25	10.25	4.1	4.1	4.1	41.76
SD		0.096	0.0961	0.0951	0.041	0.054	0.039	0.738
P9	90°	10.05	10.05	10.05	4.10	4.00	4.05	40.70
P10		10.10	10.00	10.05	4.00	4.00	4.00	40.20
P11		10.10	10.05	10.075	4.00	4.00	4.00	40.30
P12		10.05	10.00	10.025	4.05	4.00	4.025	40.35
Min		10.05	10	10.025	4	4	4	40.2
Max		10.1	10.05	10.075	4.1	4	4.05	40.7
SD		0.041	0.027	0.028	0.044	0	0.022	0.255
S1	0°	10.25	10.25	10.25	4.15	4.10	4.125	42.28
S2		10.50	10.30	10.40	4.10	4.05	4.075	42.38
S3		10.30	10.30	10.30	4.15	4.10	4.125	42.48
S4		10.15	10.25	10.20	4.10	4.05	4.075	41.56
Min		10.15	10.25	10.2	4.1	4.05	4.075	41.56
Max		10.5	10.3	10.4	4.15	4.1	4.125	42.48
SD		0.185	0.125	0.148	0.061	0.041	0.051	1.037
S5	45°	10.55	10.45	10.50	4.25	4.30	4.275	44.88
S6		10.15	10.25	10.20	4.10	4.15	4.125	42.07
S7		10.25	10.30	10.275	4.30	4.30	4.30	44.18
S8		10.35	10.30	10.325	4.35	4.35	4.35	44.91
Min		10.15	10.25	10.2	4.1	4.15	4.125	42.07
Max		10.55	10.45	10.5	4.35	4.35	4.35	44.91
SD		0.207	0.163	0.182	0.145	0.1443	0.144	2.134
S9	90°	10.10	10.10	10.10	4.25	4.25	4.25	42.92
S10		10.10	10.20	10.15	4.20	4.10	4.15	42.12
S11		10.10	10.05	10.075	4.10	4.10	4.10	41.30
S12		10.10	10.10	10.10	4.30	4.25	4.275	43.17
Min		10.1	10.05	10.075	4.1	4.1	4.1	41.3
Max		10.1	10.2	10.15	4.3	4.25	4.275	43.17
SD		0.044	0.074	0.054	0.120	0.108	0.112	1.291

**Table 3 polymers-15-01113-t003:** Main geometric dimensions of the compression test specimens and the standard deviations after printing each printing layer orientation.

Specimens	Layer Orientation	Mesurement 1	Mesurement 2	Average ø [mm]	Mesurement 2	Cross-Section Area [mm^2^]
		ø [mm]	ø [mm]		Hight [mm]	
1	90°	12.30	12.30	12.30	25.50	118.76
2		12.40	12.70	12.55	25.30	123.63
3		12.30	12.60	12.45	25.30	121.67
4		12.50	12.50	12.50	25.50	122.65
Min		12.3	12.3	12.3	25.3	118.76
Max		12.5	12.7	12.55	25.5	123.63
SD		0.1675	0.167	0.145	0.1	1.880
5	45°	12.35	12.60	12.475	25.55	122.16
6		12.50	12.45	12.475	26.00	122.16
7		12.60	12.60	12.60	25.25	124.62
8		12.20	12.20	12.20	25.70	116.83
Min		12.2	12.2	12.2	25.25	116.83
Max		12.6	12.6	12.6	26	124.62
SD		0.198	0.194	0.187	0.288	2.902
9	0°	12.35	12.30	12.325	26.00	119.24
10		12.50	12.30	12.40	25.65	120.70
11		12.30	12.30	12.30	26.00	118.76
12		12.45	12.45	12.45	25.95	121.67
Min		12.3	12.3	12.3	25.65	118.76
Max		12.5	12.45	12.45	26	121.67
SD		0.155	0.174	0.159	0.266	1.649

## Data Availability

The data presented in this study are available upon request from the corresponding author.

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
