# Peer review of "The Influence of Printing Layer Thickness and Orientation on the Mechanical Properties of DLP 3D-Printed Dental Resin"

_polymers, 2023, doi:10.3390/polym15051113_

Round 1

Reviewer 1 Report

The present paper addresses a topic of great interest to both researchers and clinicians, but the research protocol used is obsolete and badly defined. Major revisions are needed.

-English revision is strongly required by a native English speaker.

-The "Introduction" section is insufficient. It is necessary to report the state of the art on the topic addressed, and what the literature says about the chosen topic. A detailed analysis of the gaps present in the literature is missing. Please, report the outcomes extracted from the literature in reference to the objectives of your study.

-It is necessary to insert at least one null hypothesis at the end of the introduction section.

-Provide manufacturer information in brackets, after mentioning each material or software used throughout the text (Version and Model, Manufacturer, Country, City).

-Figure 4: Photo quality is poor. The images are confusing. Please provide a better Figure 4.

-Was each experiment conducted under the same environmental conditions of temperature, relative humidity, and air pressure? If so, please provide information about it in the manuscript. The same environmental conditions are relevant in order to avoid the risk of bias error.

-Was a randomization process made and which method was used?

-Was a power analysis made to determine the study's sample size? If so, please include this information. If not, please explain why not and state how sample sizes were determined.

-It is important to clarify if tests to evaluate the presence of a normal distribution (i.e., Shapiro Wilk, Kolmogorov-Smirnov) and homogeneity of the variances (i.e., Levene) were run. Besides, their results should be reported in the Results section.

-The "Discussion" section should begin by stating whether the null hypothesis has been accepted or rejected.

-A detailed paragraph regarding the limitations of this clinical study is mandatory in the "Discussion" section.

The "Conclusions" section is excessively long and distracting. It is requested to reduce it and make it more concise.

-The list of References is scarce and many recent papers are missing. Please update the reference list.

Author Response

Dear reviewer 1,

Thank you for the time to follow our manuscript. Also, we thank you for your recommendations to improve the manuscript. The revised manuscript was corrected according to your observation.

The present paper addresses a topic of great interest to both researchers and clinicians, but the research protocol used is obsolete and badly defined. Major revisions are needed.

Comment 1: -English revision is strongly required by a native English speaker.

Answer 1: Thank you for the observation, we agree. Proper English proofing was undergone

Comment 2: -The "Introduction" section is insufficient. It is necessary to report the state of the art on the topic addressed, and what the literature says about the chosen topic. A detailed analysis of the gaps present in the literature is missing. Please, report the outcomes extracted from the literature in reference to the objectives of your study.

Answer 2: We expanded the introduction section, emphasizing the state of the art and clearly pointed out the gap in knowledge that our research intended to fill. Thank you for the observation, your comment was verry helpful in our revision process.

Comment 3: -It is necessary to insert at least one null hypothesis at the end of the introduction section.

Answer 3: We have better enunciated the aim of the present study, highlighting all of the objectives that we were hoping to achieve experimentally.

Comment 4: -Provide manufacturer information in brackets, after mentioning each material or software used throughout the text (Version and Model, Manufacturer, Country, City).

Answer 4: Thank you for the comment, the issue was corrected.

Comment 5: -Figure 4: Photo quality is poor. The images are confusing. Please provide a better Figure 4.

Answer 5: The issue was corrected. During the revision process figure 4 became figure 3. We hope that the quality is good enough for proper understanding of the images.

Comment 6: -Was each experiment conducted under the same environmental conditions of temperature, relative humidity, and air pressure? If so, please provide information about it in the manuscript. The same environmental conditions are relevant in order to avoid the risk of bias error.

Answer 6: Yes, each experiment was conducted under the same environmental conditions of temperature, relative humidity, and air pressure. The information has now been specified in the text.

Comment 7 -Was a randomization process made and which method was used?

Answer 7: The samples for each mechanical test were modeled according to the standards in the field and printed at the highest resolution possible (with the equipment mentioned in the manuscript) all at once to benefit the same printing conditions. No randomization process was considered relevant.

Comment 8: -Was a power analysis made to determine the study's sample size? If so, please include this information. If not, please explain why not and state how sample sizes were determined.

Answer 8: Sample size was decided after analysis of literature on experimental studies done for tensile strength and compression strength determination. The revised material and methods section better addresses this subject.

Comment 9: -It is important to clarify if tests to evaluate the presence of a normal distribution (i.e., Shapiro Wilk, Kolmogorov-Smirnov) and homogeneity of the variances (i.e., Levene) were run. Besides, their results should be reported in the Results section.

Answer 9: The statistical tests that was included in our research is now described in the results section.

Comment 10: -The "Discussion" section should begin by stating whether the null hypothesis has been accepted or rejected.

Answer 10: Thank you for this comment, it helped us in the revision of the whole discussion section, we hope that in the present form it is more suitable for understanding.

Comment 11: -A detailed paragraph regarding the limitations of this clinical study is mandatory in the "Discussion" section.

The "Conclusions" section is excessively long and distracting. It is requested to reduce it and make it more concise.

Answer 11: We have revised both discussion and conclusions sections of this experimental mechanical tests study. Thank you for your recommendations.

Comment 12 -The list of References is scarce and many recent papers are missing. Please update the reference list.

Answer 12: In the revised version of the manuscript the reference list was updated.

Best regards,

The Authors

Reviewer 2 Report

The authors have experimentally determined the mechanical characteristics of the biocompatible resin from NextDent C&B Micro-Filled Hybrid (MFH), a 3D printing resin used in the field of dental medicine and dental laboratories, using tensile and compression testing.

The novelty of the work is not clear and the manuscript is not well-written. It needs to be revised and rewritten carefully. The paper in its current form cannot be accepted for publication in the journal of polymers.

Here are some minor and major comments and points that require to be considered.

Do not use the terms "we", "us", and so on in scientific work particularly in the abstract.

The first sentence of the abstract says you want to measure the mechanical properties of "NextDent C&B Micro-Filled Hybrid (MFH)" while you already have them via the material data sheet. Revise this or explain.

Page 1, line 31: Name some of the key topics for the functioning and development of human society.

Page 1 line 35: "These advances also had a considerable impact in dental medicine". This statement doesn't need three references.

Are you sure that you can't find the tensile strength of the resin in its technical sheet?

The second half of the introduction is focused on FEM, while it is not relevant to the work done in this paper.

The last paragraph of the introduction must clearly cover the novelty of the work.

As mentioned above, the FEM is not relevant to this work. If it is removed from the introduction, this section will be too short. Totally, the introduction section is too poor and should be improved.

In the material section, you need to mention where you have provided the resin (company name, country, product code, etc.)

The printing directions are not clear. If according to figure 1, you have printed them in 0, 45, and 90 with respect to axis y, a challenging question is why you did not print them in the x-y plane and change the printing directions in this plane?

Following the previous comment, why did you print them in the direction of the width of dog-bone test samples? Explain.

Use a schematic figure to show the printing directions.

Page 4, line 103: "The dimensions of all specimens were measured with an electronic caliper according to precision standards [25]" does it really need a reference?

The data in table 2 doesn't need to be here. What are 1.1 and 1.2 in table 2? The data for the calculation of cross-section areas of different samples is not necessary to be included in the main body of the paper. The same comment for table 3.

Page 4, line 104: "Tensile specimens were centralized according to the orientation of the polymerization/printing layers in Table 2. " what does this mean? It is not clear.

Page 4, line 120: The main characterizations of Zwick/Roell Z005 universal machine don't need to be in a research paper. You must mention the exact test conditions such as speed, load cell, etc. of the tensile test conducted.

Page 5, line 126: This sentence doesn't need 3 references.

The paper is not like a research paper. The manuscript must be rewritten carefully.

The test conditions for both tensile and compression should be added to the manuscript.

Page 5, line 142: "The colors of the curves are based on the orientation of the print layer with respect to the printer tray and are marked as follows" what curves? Mention them properly.

Page 6, line 162: what is the meaning of "the maximum stress-strain values"? Stress and strain are two different mechanical properties and should be compared separately for different samples.

Page 7, lines 176-180: we already knew this conclusion. Mention the percentages to represent the changes and differences.

The fracture behaviour of tested samples is discussed while there is no figures to support the discussion.

Page 7, line 192: "The maximum values on the stress-strain curves are recorded among the specimens that were printed with a direction of the print layer..." what is the difference between this and what you mention in the first sentences of the "discussion" section? It is repeated.

The discussion section is more similar to a report than a scientific discussion. You must discuss your results not just report them.

The main achievements of the work should be summarised by bullets in the conclusion section.

Totally, the achievements of the work are not clear.

Author Response

Dear reviewer 2,

Thank you for your time to follow our manuscript. Also, we thank you for your recommendations to improve the manuscript. The revised manuscript was corrected according to your observation.

The authors have experimentally determined the mechanical characteristics of the biocompatible resin from NextDent C&B Micro-Filled Hybrid (MFH), a 3D printing resin used in the field of dental medicine and dental laboratories, using tensile and compression testing.

The novelty of the work is not clear and the manuscript is not well-written. It needs to be revised and rewritten carefully. The paper in its current form cannot be accepted for publication in the journal of polymers.

Here are some minor and major comments and points that require to be considered.

Comment 1: Do not use the terms "we", "us", and so on in scientific work particularly in the abstract.

Answer 1: Thank you for your observation, the issue was corrected.

Comment 2: The first sentence of the abstract says you want to measure the mechanical properties of "NextDent C&B Micro-Filled Hybrid (MFH)" while you already have them via the material data sheet. Revise this or explain.

Answer 2: This research sought to determine the properties of the resin while subjected to different types of loads, like tensile loads (to determine the Young’s Modulus for tensile strength) or under a compressive load (to determine the Compression Modulus). As mentioned in the text, the material data sheet only provides an approximate value for the resin’s Flexural strength (≥ 50 MPa = 107). In the revised manuscript we have highlighted this gap in knowledge and why the findings of the present experimental study would be relevant

Comment 3: Page 1, line 31: Name some of the key topics for the functioning and development of human society.

Answer 3: The introduction section was completely revised. We hope that in the present form the state of the art is better described.

Comment 4: Page 1 line 35: "These advances also had a considerable impact in dental medicine". This statement doesn't need three references.

Answer 4: Thank you for the observation, the issue was corrected

Comment 5: Are you sure that you can't find the tensile strength of the resin in its technical sheet?

Answer 5: No other mechanical properties besides the flexural strength can be found in the manufacturers technical sheet or in literature. The present study was aimed at filling a gap in the knowledge about the resin material used in order to facilitate future studies.

Comment 6: The second half of the introduction is focused on FEM, while it is not relevant to the work done in this paper.

Answer 6: The introduction section was completely revised. Thank you for your observation

Comment 7: The last paragraph of the introduction must clearly cover the novelty of the work.

Answer 7: In the revised version of the manuscript, we focused on better emphasizing the aim and importance of the present study. We hope that you find the new version better suited for publication.

Comment 8: As mentioned above, the FEM is not relevant to this work. If it is removed from the introduction, this section will be too short. Totally, the introduction section is too poor and should be improved.

Answer 8: The introduction section was completely revised. Thank you for your observation.

Comment 9: In the material section, you need to mention where you have provided the resin (company name, country, product code, etc.)

Answer 9: The data of origin of the material were specified in the text.

Comment 10: The printing directions are not clear. If according to figure 1, you have printed them in 0, 45, and 90 with respect to axis y, a challenging question is why you did not print them in the x-y plane and change the printing directions in this plane?

Following the previous comment, why did you print them in the direction of the width of dog-bone test samples? Explain.

Use a schematic figure to show the printing directions.

Answer 10: Figure 1 was modified to better highlight the printing layer orientations, and comments were added both in the figure caption and in the material and method section to describe the angles. The direction of the 3D printer tray and printing head (in this case the digital projector screen of the printer) is fixed, the only way to obtain different orientations of the printing layers is to rotate the model at the required angle before creating the 3D printing G-code. Considering these limitations, tilting the models on the x-y plane would have led to the same direction of printing for all the samples. Direction of the layers that also needed to be different to the direction of the load applied by the universal tensile/compression machines in order to have relevance.

Comment 11: Page 4, line 103: "The dimensions of all specimens were measured with an electronic caliper according to precision standards [25]" does it really need a reference?

The data in table 2 doesn't need to be here. What are 1.1 and 1.2 in table 2? The data for the calculation of cross-section areas of different samples is not necessary to be included in the main body of the paper. The same comment for table 3.

Answer 11: Considering that we mentioned a standardized method, we thought it is best to cite it in text, the reference was removed. The tables were corrected. Thank you for the observation.

Comment 12: Page 4, line 104: "Tensile specimens were centralized according to the orientation of the polymerization/printing layers in Table 2. " what does this mean? It is not clear.

Answer 12: The phrase was corrected. We meant to say dimensions of the specimens printed for tensile strength testing. Thank you for the observation.

Comment 13: Page 4, line 120: The main characterizations of Zwick/Roell Z005 universal machine don't need to be in a research paper. You must mention the exact test test conditions such as speed, load cell, etc. of the tensile conducted.

Answer 13: The issue was addressed, thank you for your comment

Comment 14: Page 5, line 126: This sentence doesn't need 3 references.

Answer 14: The issue was addressed, thank you for your comment

Comment 15: The paper is not like a research paper. The manuscript must be rewritten carefully.

Answer 15: We have carefully revised the whole manuscript taking into consideration reviewer observations and comments, and hope that in the present form you find it more suitable for publication.

Comment 16: The test conditions for both tensile and compression should be added to the manuscript.

Answer 16: The test conditions are now specified in the manuscript. Thank you for pointing out this issue.

Comment 17: Page 5, line 142: "The colors of the curves are based on the orientation of the print layer with respect to the printer tray and are marked as follows" what curves? Mention them properly.

Answer 17: in the revised version we addressed this issue and we mentioned them properly.

Comment 18: Page 6, line 162: what is the meaning of "the maximum stress-strain values"? Stress and strain are two different mechanical properties and should be compared separately for different samples.

Answer 18: we refer to the maximum stress recorded, respectively, to the maximum strain recorded, but taken separately.

Comment 19: Page 7, lines 176-180: we already knew this conclusion. Mention the percentages to represent the changes and differences.

Answer 19: In the revised version of the manuscript the whole discussion and conclusion sections were reorganized

Comment 20: The fracture behaviour of tested samples is discussed while there is no figures to support the discussion.

Answer 20: We discussed the fracture behavior of the specimens based on Figure 4.

Comment 21: Page 7, line 192: "The maximum values on the stress-strain curves are recorded among the specimens that were printed with a direction of the print layer..." what is the difference between this and what you mention in the first sentences of the "discussion" section? It is repeated.

Answer 21: In the revised version of the manuscript the whole discussion and conclusion sections were reorganized

Comment 22: The discussion section is more similar to a report than a scientific discussion. You must discuss your results not just report them.

Answer 22: In the revised version of the manuscript the whole discussion and conclusion sections were reorganized

Comment 23: The main achievements of the work should be summarised by bullets in the conclusion section.

Answer 23: The conclusion section is now reorganized and the main achievements of the work are better highlighted

Comment 24: Totally, the achievements of the work are not clear.

Answer 24: The conclusion section is now reorganized and the main achievements of the work are better highlighted

Best regards,

The Authors

Round 2

Reviewer 1 Report

I am happy with the changes made by the Authors after my comments.

Reviewer 2 Report

The authors have addressed the questions and comments properly, so the paper is accepted in its current form.